# Microalgae as an Efficient Vehicle for the Production and Targeted Delivery of Therapeutic Glycoproteins against SARS-CoV-2 Variants

**DOI:** 10.3390/md20110657

**Published:** 2022-10-23

**Authors:** Jaber Dehghani, Ali Movafeghi, Elodie Mathieu-Rivet, Narimane Mati-Baouche, Sébastien Calbo, Patrice Lerouge, Muriel Bardor

**Affiliations:** 1Université de Rouen Normandie, Laboratoire GlycoMEV UR 4358, SFR Normandie Végétal FED 4277, Innovation Chimie Carnot, F-76000 Rouen, France; 2Department of Plant, Cell and Molecular Biology, Faculty of Natural Sciences, University of Tabriz, Tabriz 5166616471, Iran; 3Université de Rouen Normandie, Inserm U1234, F-76000 Rouen, France

**Keywords:** ACE2, biopharmaceuticals, edible vaccine, COVID-19, microalgae, SARS-CoV-2, S-glycoprotein, synthetic biology, targeted delivery

## Abstract

Severe acute respiratory syndrome–Coronavirus 2 (SARS-CoV-2) can infect various human organs, including the respiratory, circulatory, nervous, and gastrointestinal ones. The virus is internalized into human cells by binding to the human angiotensin-converting enzyme 2 (ACE2) receptor through its spike protein (S-glycoprotein). As S-glycoprotein is required for the attachment and entry into the human target cells, it is the primary mediator of SARS-CoV-2 infectivity. Currently, this glycoprotein has received considerable attention as a key component for the development of antiviral vaccines or biologics against SARS-CoV-2. Moreover, since the ACE2 receptor constitutes the main entry route for the SARS-CoV-2 virus, its soluble form could be considered as a promising approach for the treatment of coronavirus disease 2019 infection (COVID-19). Both S-glycoprotein and ACE2 are highly glycosylated molecules containing 22 and 7 consensus *N*-glycosylation sites, respectively. The *N*-glycan structures attached to these specific sites are required for the folding, conformation, recycling, and biological activity of both glycoproteins. Thus far, recombinant S-glycoprotein and ACE2 have been produced primarily in mammalian cells, which is an expensive process. Therefore, benefiting from a cheaper cell-based biofactory would be a good value added to the development of cost-effective recombinant vaccines and biopharmaceuticals directed against COVID-19. To this end, efficient protein synthesis machinery and the ability to properly impose post-translational modifications make microalgae an eco-friendly platform for the production of pharmaceutical glycoproteins. Notably, several microalgae (e.g., *Chlamydomonas reinhardtii*, *Dunaliella bardawil*, and *Chlorella* species) are already approved by the U.S. Food and Drug Administration (FDA) as safe human food. Because microalgal cells contain a rigid cell wall that could act as a natural encapsulation to protect the recombinant proteins from the aggressive environment of the stomach, this feature could be used for the rapid production and edible targeted delivery of S-glycoprotein and soluble ACE2 for the treatment/inhibition of SARS-CoV-2. Herein, we have reviewed the pathogenesis mechanism of SARS-CoV-2 and then highlighted the potential of microalgae for the treatment/inhibition of COVID-19 infection.

## 1. Introduction

To date, people around the world have been impacted by the recently emerged Coronavirus Disease 2019 (COVID-19) first reported in Wuhan, China, in late December 2019 [1]. This infectious disease is caused by severe acute respiratory syndrome coronavirus 2 (SARS-CoV-2) and as of October 2022, over 617 million cases have been detected in about 228 countries, with about 6.5 million deaths [2]. The rapid and efficient person-to-person transmission of SARS-CoV-2 led the World Health Organization (WHO) to declare COVID-19 infection a pandemic [3]. SARS-CoV-2 is an 80–120 nm long positive single-stranded RNA virus that is the third identified coronavirus causing severe pneumonia in humans after SARS-CoV-1 and Middle East respiratory syndrome coronavirus (MERS-CoV). Despite a lower mortality rate (approximately 1–2%), SARS-CoV-2 has shown a higher capacity for contagion and infection than SARS-CoV-1 and MERS-CoV [4].

SARS-CoV-2 mainly targets the respiratory tract, resulting in rapid and severe respiratory symptoms and lung failure, as well as some clinical symptoms such as fever, dry cough, fatigue, and dyspnea [5]. In addition, studies reported gastrointestinal disturbances such as loss of appetite followed by diarrhea, nausea, and abdominal pain in the infected patients [6]. Indeed, SARS-CoV-2 can also acutely replicate in the mucosa of the patient’s small intestine and excrete its RNA into the patient’s stool [7]. In addition to these manifestations, many patients have exhibited a variety of symptoms (e.g., olfactory and gustatory disturbances, anosmia, headache, dysgeusia, confusion, and fatigue), which could be attributed to cranial nerve involvement [8]. SARS-CoV-2, as with other coronaviruses, may initially invade peripheral-nerve endings and then progress regularly to the central nervous system via synaptic-connected junctions [9].

Many drugs are currently evaluated for the treatment or prevention of COVID-19 [10], meanwhile many researchers have attempted to construct an effective vaccine against COVID-19. Subsequently, several types of vaccines (i.e., attenuated, mRNA, and recombinant vaccines) have been urgently approved [1,11]. Most of the available SARS-CoV-2 vaccines, drugs, and monoclonal antibodies (mAbs) target the S-glycoprotein. For example, the approved mRNA vaccines (Moderna/US NIAID and Pfizer-BioNTech) contain the sequences for the synthesis of the stabilized prefusion forms of the full-length S-glycoprotein of SARS-CoV-2. After administration, mRNAs use host cells’ transcription and translation machineries to produce the SARS-CoV-2 S-glycoprotein. Subsequently, the expressed viral antigen is recognized by T- and B-lymphocytes in host cells, triggering adaptive immune responses directed against the virus [12]. In addition, vaccines based on replication-defective viral vectors have been used in human adenovirus serotype 5 (e.g., Ad5-nCoV) or chimpanzee adenovirus (i.e., AZD1222) platforms to deliver the full-length S-glycoprotein coding sequence of SARS-CoV-2 into human host cells. Approved protein-based subunit vaccines against the SARS-CoV-2 are also composed of the full-length S-glycoprotein (e.g., NVX-CoV2373) or its fragments (e.g., ZF2001) [13]. Recently, a virus-like particle vaccine (i.e., Medicago Covifenz) produced in tobacco plants, *Nicotiana benthamiana*, was successfully approved in Canada against COVID-19 infection. This type of vaccine contains a trimeric form of the S-glycoprotein SARS-CoV-2 that is assembled on the surface of the virus-like particle to trigger the immune response in the host cells [14]. Of note, several monoclonal antibodies (mAbs), such as imdevimab, casirivimab, and sotrovimab targeting the S-glycoprotein, were also developed to inhibit SARS-CoV-2 infection [15].

However, these therapeutic approaches have progressively lost their efficacy in inhibiting new variants of SARS-CoV-2. In fact, SARS-CoV-2 continuously mutates due to its high mutational rate as an RNA virus, shorter replication time, and low genome stability [16]. These emerging variants generally exhibit a more rapid ability to spread and an easier adaptation process to new environmental conditions, resulting in vaccine resistance and the ability to evade the host immune system [16,17]. Most of the mutations occurred in different regions of the SARS-CoV-2 S-glycoprotein, which is targeted for the construction of vaccines and biologics directed against COVID-19 [18]. In the Omicron variant, for example, nearly 26–32 mutations occurred in the S-glycoprotein, including approximately 15 mutations located in the receptor-binding domain (RBD) region [19]. Unfortunately, these S-glycoprotein mutations significantly enhance the transmissibility and immune system evasion capacity of several new emerging SARS-CoV-2 variants, such that the WHO has declared some of them (e.g., Omicron) as variants of concern [17]. However, regardless of the S-glycoprotein mutations, all SARS-CoV-2 variants bind to the ACE2 receptor to enter human cells. In this sense, soluble ACE2 therapy might be considered as a more effective process for the treatment of SARS-CoV-2 [20]. Therefore, it is essential to produce this glycoprotein in heterologous systems and then deliver it to target cells as a decoy to bind to SARS-CoV-2, resulting in virus inactivation [19].

Today, most approved vaccines or biologics against COVID-19 infection are primarily produced in mammalian cell lines, especially Chinese hamster ovary (CHO) cells, or more recently in plant cells (i.e., *N. benthamiana*). However, the complexity and high cost of media, low growth rate, and sensitivity to stress and pathogens have significantly increased the costs of producing biologics through the mammalian platform. Slow growth rates, possible environmental contamination or gene flowing, and rapid degradation of the produced recombinant proteins are the main concerns regarding the production of biologics by the plant systems [21,22]. Therefore, the exploration of new eco-friendly and efficient biosystems, such as microalgae, for the production of therapeutics biologics is highly demanded.

Microalgae are eukaryotic, microscopic, and photosynthetic lower organisms that have recently been considered a more promising platform for the production of various biologics, especially complex glycosylated proteins. Here, we describe the biology, physiology, and pathobiology of the SARS-CoV-2 virus to highlight the potential of microalgae for the production of recombinant SARS-CoV-2 S-glycoprotein and soluble human ACE2. Since some microalgae have already been approved as human foods by the FDA, we also described an original concept for the treatment of COVID-19 patients using microalgae for the edible delivery of the produced recombinant S and ACE2 glycoproteins.

## 2. SARS-CoV-2 Virus and Its Pathogenic Mechanism

Phylogenetic analyses have revealed that SARS-CoV-2 belongs to the β-coronaviruses lineage B, where it shows more than 80% genetic similarity to the previously reported SARS-CoV-1 [23]. In addition, amino acid sequences of S-glycoproteins of both viruses share approximately 76% identity [24]. The S-glycoprotein of SARS-CoV-2 is formed by homotrimers exposed to the viral surface, playing a key role in the virus entry into human cells [25]. Each monomer contains approximately 1273 amino acids consisting of an *N*-terminal signal peptide (1–13 residues), S1 (14–685 residues), and S2 subunits (686–1273 residues) (Figure 1). The S1 subunit is responsible for binding the virus to its host cellular receptor, while the S2 subunit allows the fusion of viral and host cellular membranes [26]. Similar to SARS-CoV-1, SARS-CoV-2 targets ACE2 as its functional cellular receptor [10,27,28]. The RBD located in the S1 subunit was also shown to interact directly with the ACE2 receptor, facilitating the pathogenesis of COVID-19 [10]. In detail, the interaction between SARS-CoV-2 S-glycoprotein and ACE2 provides the virus entry into human cells through priming of the S-glycoprotein mediated by the host transmembrane serine proteases such as transmembrane protease serine 2 (TMPRSS2), host furin-like enzymes, and cathepsin B/L in some cases (Figure 1) [29]. Bioinformatic approaches confirmed that the 3-dimensional structures of the RBD of SARS-CoV-2 were almost identical to those reported for other coronaviruses [27]. Towards this direction, it has been reported that the Glu-394 in the RBD of SARS-CoV-2 corresponds to the Asn-479 in the RBD of SARS-CoV-1. Both residues might be involved in the recognition mechanism of the human ACE2 receptor since they have been found in the hot spot of the ACE2 Lys-31 [30,31]. However, based on cryo-electron microscopy data, it has been reported that the SARS-CoV-2 S-glycoprotein can recognize and bind to the human ACE2 receptor with ~20 times higher affinity in comparison with SARS-CoV-1 [32]. Moreover, it should be noted that ongoing mutations that occurred in different parts of S-glycoprotein generated new variants with higher affinity for the human ACE2 receptor. The Omicron variant, for example, exhibits a stronger Coulomb force between its S-glycoprotein and human ACE2 receptor, resulting in significantly increased electric charges in the nucleocapsid proteins [33]. These facts indicate that targeting both S-glycoprotein and ACE2 through their recombinant counterparts may be effective in the inhibition/controlling of COVID-19 infection [34,35].

## 3. Therapeutic Potential of SARS-CoV-2 S-Glycoprotein

As S-glycoprotein plays an unavoidable role in the SARS-CoV-2 entry into human cells, it represents the main target for vaccine design and biologics development against COVID-19 infection [13]. Once the SARS-CoV-2 was imported into the host cells, the human immune system mainly recognized its surface epitopes located in S-glycoprotein resulted in innate and adaptive immune responses [36]. These immune reactions are contributed mainly by Toll-like receptors (TLR3, 7, and 8) causing enhanced production of interferon. After the replication of the virus within the lung cells, cytokines (IL-6, IL-8, IL-10, and TNF-a) and chemokines were secreted by alveolar macrophages which functioned as inflammatory mediators. Simultaneously, neutrophils, natural killer cells, monocytes, CD4+, and CD8+ T cells initiated the phagocytosis of infected cells [37]. Moreover, whole S-glycoprotein or its immunogenic fragments (amino acids from 100–280, 430–590, and 1060–1150) can trigger long-lasting dominant neutralizing immune cells against SARS-CoV-2 [11]. As a consequence, antibodies against recombinant whole or specific fragments of S-glycoprotein could block virus entry and also inhibit viral replication [38]. The S1 subunit is a receptor-binding element that contains an *N*-terminal signal peptide, an *N*-terminal domain (so-called NTD), and the RBD domain. While the S2 subunit is composed of a fusion peptide domain (FP), two heptad-repeat domains (HR1 and HR2), a transmembrane domain, and a *C*-terminal domain that functions in the fusion of viral and host membranes facilitating virus entry (Figure 1) [25]. Notably, S-glycoprotein is highly glycosylated with 22 *N*-glycosylation sites and 17 *O*-glycosylation sites. Thus, the glycosylation processes are crucial for the correct folding and biological activity of S-glycoprotein. Moreover, the glycosylation profile of S-glycoprotein is critical for host recognition, binding, penetration, and pathogenesis of the SARS-CoV-2 virus [39]. The S-glycoprotein plays a critical role in the initiation of immune responses such as secretion of neutralizing-antibody, and T-cell-related reactions during SARS-CoV-2 infection [26]. In this context, this glycoprotein is considered as the pivotal antigenic molecule that leads to the production of immunity-related elements such as neutralizing mAbs against virus infection [13]. Initially, it was reported that antibodies produced during COVID-19 infection are enabled to bind to the RBD of both SARS-CoV-1 and SARS-CoV-2, suggesting that this region would undergo less mutation [13]. Later on, this theory was rejected as new SARS-CoV-2 variants carrying a different number of insertion or deletion mutations in their S-glycoproteins, even in the RBD region, have emerged [19]. Thus far, the potential of the entire S-glycoprotein and its antigenic elements, such as the S1 subunit through the RBD domains, have been evaluated as candidates for SARS-CoV-2 vaccine construction. Although full-length S-glycoprotein-related vaccines can induce potent neutralizing antibodies and other immune responses, some reports have shown that the secreted antibodies may enhance viral infection after reinfection with a SARS-CoV-2 homologous or cause liver damage [40,41].

As an alternative approach, immunogenic fragments of S-glycoprotein could be targeted for the production of subunit vaccines against SARS-CoV-2. For example, vaccines based on the S1-related subunit could significantly elicit the production of neutralizing antibodies providing strong protective immunity against viral infection. Of note, the S2 fragments had lower immunogenicity, resulting in lower antibody induction and fewer immune responses against viral infection [40]. In addition to vaccine construction, S-glycoprotein has been regularly used for the construction of diagnostic kits for infected patients with COVID-19 [42]. Indeed, most molecular diagnosis kits (RT-PCR) against SARS-CoV-2 are designed using the S1 subunit, as it has less cross-reactivity than the full-length S-glycoprotein, less similarity to human-related coronaviruses, and more immunogenicity than the S2 subunit [43,44].

## 4. ACE2 Is Also a Promising Approach for the Treatment of COVID-19 Infection

The ACE2 is a type 1 membrane glycoprotein that is an enzymatically active homolog of ACE. ACE is the major component of the renin-angiotensin-aldosterone system (RAAS), involved in blood pressure regulation, electrolyte, fluid homeostasis, natriuresis, and salt balance [45]. In detail, angiotensinogen produced in the liver is cleaved by renin, resulting in the formation of angiotensin I (Ang I) decapeptide. Subsequently, ACE protein converts Ang I to the octapeptide Ang II, which imposes its physiological function mainly including blood pressure increase, vasoconstriction, aldosterone synthesis, and induction of inflammatory and pro-fibrotic pathways through the angiotensin II type I receptor (AT1R) [10,46,47]. Despite the similarity of more than 61% in the amino acid sequences of their catalytic domains, several differences exist between ACE and ACE2. ACE is a dipeptidyl-carboxypeptidase enzyme with two zinc-binding motifs in its *N*- and *C*-terminal domains, whereas ACE2 is a mono-carboxypeptidase with one zinc-binding motif in its *N*-terminus domain [48]. Therefore, ACE2 could act as a negative regulator of RAAS pathways in which it hydrolyzes Ang II into the Ang 1–7 inducing anti-inflammatory, anti-fibrotic, and anti-proliferation cascades via its binding to the Mas receptor (MasR) [10,49]. Indeed, ACE2 actively counteracts the physiological effects of ACE, whereby disruption of the ACE/ACE2 balance in humans can cause several intense disorders (Figure 2) [50,51]. It should be noted that ACE2 is expressed in its functional form in most vital tissues, especially in healthy human organs such as the lung, skin, oral and nasal mucosa, nasopharynx, gastrointestinal system, liver, kidney, and brain tissues [52]. Importantly, the highest surface expression of ACE2 protein is reported in alveolar epithelial cells of the lung and enterocytes of the small intestine, where human cells are in contact with the external environment, facilitating the entry of pathogens such as SARS-CoV-2 [27].

The human ACE2 is composed of three regions, namely, extracellular, transmembrane, and intracellular segments, among which the extracellular part contains a peptidase domain that participates in binding to the SARS-CoV-2 RBD [53]. The binding of S-glycoprotein to ACE2 is the initial step in the entry, replication, and subsequent pathogenesis of SARS-CoV-2. In vitro and in vivo studies revealed that ACE2 is the only functional receptor for SARS-CoV-2 entry into human cells (Figure 1) [27]. The overexpression of ACE2 originated from humans, Chinese horseshoe bats, civets, and pigs in the HeLa cells allowed SARS-CoV-2 infection, confirming that the virus uses ACE2 as an entry receptor [54]. Further data showed that SARS-CoV-2 could not use other coronavirus receptors such as the aminopeptidase N and dipeptidyl peptidase 4 [27,54]. It was also previously shown that mice infected with SARS-CoV-1 had a significant reduction in ACE2 expression in their lungs [28]. Given the high similarity between SARS-CoV-1 and SARS-CoV-2, it could be hypothesized that decreased ACE2 expression levels may play an important role in the pathogenesis of both viruses. Notably, overexpression of human ACE2 can significantly increase the severity of disease in SARS-CoV-infected mice [55]. Thus, blocking this interaction could therefore logically be considered as a promising approach for the inhibition of COVID-19 infection. The emergence of new variants of SARS-CoV-2, combined with concerns about the ability to evade detection by the immune system and reluctance to be vaccinated, may diminish the success of the ongoing efforts to vaccinate the world’s populations against COVID-19. Consequently, soluble ACE2 therapy could be designated as a new promising strategy for the treatment of infection [56].

In addition, injection of SARS-CoV-1 S-glycoprotein into infected mice has been shown to worsen the disease by blocking the RAAS pathways [46]. Therefore, it appears that ACE2 may simultaneously serve as an entry receptor for SARS-CoV-2 and act as a protective element against the virus [57]. Hence, ACE2 has attracted the attention of many investigators as an important therapeutic molecule for the treatment of COVID-19 through different approaches. Since priming of S-glycoprotein by TMPRSS2 is required for SARS-CoV-2 entry into host cells, some serine protease inhibitors have been tested to block viral entry. For example, camostat mesylate, a potent serine protease inhibitor, can partially prevent SARS-CoV-2 entry into lung cells [58]. Interrupting the interaction between ACE2 catalytic motifs and SARS-CoV-2 S-glycoprotein by antibodies or small molecules is another approach for the treatment of patients with COVID-19 [27]. Regarding the higher binding affinity of SARS-CoV-2 to ACE2, delivery of a soluble form of ACE2 could be a potential practice for the treatment of COVID-19 patients [30]. Indeed, delivery of excessive amounts of soluble ACE2 could be used as a decoy to competitively bind to SARS-CoV-2, thereby significantly reducing viral entry into target cells [59]. This approach can rescue cellular ACE2 activity, on which it negatively regulates the RAAS to protect the lung from injury [57].

Previously, a recombinant form of human ACE2 (rhACE2) was produced and reported to be safe without causing negative hemodynamic changes in volunteers [60]. The further results from phase II clinical trials showed that infusion of rhACE2 rapidly decreased the Ang II levels, while Ang 1–7 and Ang 1–5 levels have increased in patients with acute respiratory distress syndrome (ARDS) [61]. The rhACE2 may have positive effects on the treatment of hypertension, heart and renal injuries, and hepatic fibrosis [27]. Recently, the potential of rhACE2 was evaluated for the treatment of patients with COVID-19. For this purpose, different concentrations of rhACE2 were mixed with SARS-CoV-2 and added to the Vero-E6 cell culture. The data showed that the presence of rhACE2 in the medium of infected cells can significantly inhibit SARS-CoV-2 infection depending on the amount of virus and the dose of rhACE2. Furthermore, it was identified that rhACE2 can effectively inhibit COVID-19 in SARS-CoV-2 infected human capillary and renal organoids where ACE2 is highly expressed. It is believed that soluble rhACE2 may act as a protective agent against lung injury or block the entry of SARS-CoV-2 into target cells [57]. However, rhACE2 therapy presented some challenges that significantly increase production and treatment costs. Given the short in vivo half-life of rhACE2, continuous infusion of the protein is required to improve the efficacy of the treatment period [59]. Recently, a chimeric rhACE2-IgG2 Fc fusion was produced, which demonstrated that this protein had higher stability and half-life in mouse plasma [62]. In addition, the rhACE2-IgG2 Fc fusion molecule could efficiently block SARS-CoV-2 entry and inhibits S-glycoprotein-mediated cell-cell fusion. Importantly, this chimeric form of ACE2 could neutralize various recently emerged strains of SARS-CoV-2 such as B.1.1.7 (Alpha), B.1.351 (Beta), B.1.617.1 (Kappa), and B.1.617.2 (Delta). In vivo data showed that rhACE2-IgG2 Fc fusion effectively protected mice from the SARS-CoV-2 infection by reduction of viral replication, inflammation, and histological changes in their lungs [63]. In another approach, lipid nanoparticles were used to deliver transcribed messenger RNA into the mammalian cells to rapidly express ACE2. Different analyses revealed that the produced ACE2 was able to bind to the RBD of SARS-CoV-2 in mouse lung cells and strongly inhibited (more than 90%) the infection with SARS-CoV-2 pseudovirus [53]. However, the production of rhACE2 in human cells is a time-consuming and expensive process. In this regard, the exploration of an environmentally-friendly platform for the production and delivery of human ACE2 is highly demanded.

## 5. Potential of Microalgae for the Production of Recombinant S-Glycoprotein and ACE2

Microalgae are unicellular photosynthetic eukaryotic organisms that have recently been identified as an efficient and environmentally-friendly alternative for the production of recombinant proteins, including complex glycoproteins [64,65,66,67,68]. Microalgae are able to properly impose post-translational modifications such as glycosylation and disulfide bridges on recombinant proteins [69]. Low biocontainment requirements, minimal immunogenic cross-reactive proteins, and no possibility of infection by mammalian viruses are other important advantages of microalgae for the production of medicinal biologics [70,71]. In addition, these microorganisms can efficiently assemble complex, large, multi-subunit proteins in fully functional forms [21]. For example, the full-length sequence of human CL4 mAb directed against hepatitis B surface antigen and also a recombinant mAb against Marburg virus nucleoprotein have been successfully expressed in the microalga *Phaeodactylum tricornutum* as a full-length stable and functional form [65,66,68,72]. Likewise, several human cytokines (e.g., erythropoietin, interferon alpha 2a, and interleukin 2), human hormones (i.e., growth hormone and VEGF-165), and other biologics (antigens and multi-epitope proteins) have been produced with appropriate functionality in green microalgae such as *Chlamydomonas reinhardtii*, *Chlorella*, and *Dunaliella* species (Table 1) [73,74,75]. Some eukaryotic microalgae (e.g., *C. reinhardtii*, *Chlorella vulgaris*, *Dunaliella salina*, and *P*. *tricornutum*), and also the cyanobacterium *Arthrospira platensis* (previously categorized as blue-green microalga) produce vital components, including essential amino acids and different groups of vitamins that are necessary to boost the immune system against various diseases including infectious diseases (e.g., COVID-19) due to their anti-inflammatory and tissue-repairing properties [76,77].

Severe inflammatory responses can be observed in COVID-19 patients due to out-of-control cytokine storm reactions. During this phenomenon, the release of some pro-inflammatory cytokines (e.g., IL-1β, IL-6, and TNF-α) and chemokines (i.e., CCL2–3 and CXCL9–10) increases significantly, resulting in hyperactive immune response and intense lung injury in COVID-19 patients [84]. Furthermore, the NF-κB signaling pathway may play a key role, associated with the JAK/STAT-3 pathway, in triggering the cytokine storm during COVID-19 infection. It has been reported that some peptides, pigments (i.e., violaxanthin), and crude carotenoid extracts from *Chlorella* species could reduce cytokine storm reactions [75]. For instance, oxylipins extracted from *C. debaryana* could reduce the overwhelming of cytokines through NF-κB-dependent inflammatory pathways [85]. Furthermore, preliminary reports have shown that the administration of carotenoids such as astaxanthin extracted from the microalga *Haematococcus pluvialis* could alleviate the risk of cytokine storm in COVID-19 patients [86]. Phycocyanin and phycocyanobilins of *A. platensis* are bioactive compounds with anti-inflammatory, antioxidant, and anti-tumor properties that presented a great potential to inhibit cytokine storms and SARS-CoV-2 polymerase activity [87,88]. Notably, *A. platensis* contains a sulfated polysaccharide known as calcium spirulan that could significantly inhibit the replication of some viruses, including influenza, HIV, and mumps [89]. As well, sulfated polysaccharides extracted from *C. stigmatophora* and *P. tricornutum* have shown proper anti-inflammatory activity [90]. These sulfated polysaccharides could be used for the treatment of COVID-19 [91].

Currently, most vaccines, biologics, and diagnosis kits containing different fragments of S-glycoprotein are produced in mammalian cells and thus are expensive [92]. This issue has to be considered in the context of COVID-19 treatment since sanitary strategies carried out to date have impacted seriously on the economy, society, and global health worldwide including in low-income developing countries [93]. It should be noted that plant vaccines against SARS-CoV-2 have recently been developed with promising results. Expression of S1 or smaller fragments in tobacco cells elicited a positive immune response in pre-clinical experiments and this research is progressing to Phase I/II clinical trials. The vaccine was claimed to be able to trigger significant immune responses with a single dose and was also stable at room temperature [94]. These plant-based vaccines showed high immunogenicity and were well-tolerated in phase I, II, and III clinical trials [95,96]. However, other reports have revealed that the plant-produced vaccine contained plant-specific *N*-glycan epitopes that could cause immunogenic or allergic reactions in humans [97,98]. Recently, the potential of the bacteria *Escherichia coli* and the yeast *Pichia pastoris* was evaluated for the production of SARS-CoV-2 RBD fragments [99,100]. Nevertheless, the absence of glycosylation machinery and formation of non-functional inclusion bodies in *E. coli*, as well as hyper-mannosylation in *P. pastoris* incredibly limited their potential for complex glycoproteins production [21,71].

Due to their rapid growth rate, simple and inexpensive media, and the ability to use autotrophic culture, microalgae represent a cost-effective platform for the production of recombinant ACE2 and S-glycoproteins [64,71,81]. More importantly, eukaryotic microalgae can correctly impose post-translational modification with high homogeneity on recombinant glycoproteins [67,101]. In addition, these microorganisms possess chaperons and chaperonins that are essential for the correct conformation of many glycoproteins without the formation of insoluble aggregates [21]. These characteristics make eukaryotic microalgae ideal biofactories for the production of S-glycoprotein and human ACE2. Given the large size and highly glycosylated contents of S-glycoprotein, the risk of protein misfolding, low protein yielding, and antibody-dependent enhancement could be significantly increased during the production and delivery processes [40].

Recently, the potential of *C. reinhardtii* for the production of both full-length S-glycoprotein and RBD region of SARS-CoV-2 has been evaluated (Table 1). The secreted form of the full-length S-glycoprotein was successfully produced in *C. reinhardtii* [80]. Further data showed that S-glycoprotein produced by the microalga could efficiently bind to the human ACE2 receptor, in which the in vitro data using 293T cell lines overexpressing hACE2 and hTMPRSS2 confirmed its proper biological activity.

In addition, microscopic analyses have shown that recombinant S-glycoprotein had no negative effects on the 293T cell lines [80]. Of note, SARS-CoV-2 RBD has been transiently expressed in the nuclear genome of *C. reinhardtii* and *C. vulgaris* via an *Agrobacterium*-related plasmid. However, the functionality of these recombinant RBDs has not been determined [102]. Furthermore, the RBD region of SARS-CoV-2 S-glycoprotein was expressed in *C. reinhardtii* either in an endoplasmic reticulum (ER) retained form, a secreted form in the media, or a localized form in the chloroplast. Western blotting data using an anti-RBD antibody revealed that the RBD retained in the ER and its secreted form have the expected size (51 kDa). In contrast, the RBD produced in the chloroplast was truncated and could not be recognized using the anti-RBD antibody. Further data confirmed that the purified version of RBD retained by the ER could bind specifically to the human ACE2 receptor with an affinity similar to that of a recombinant RBD expressed in mammalian cells [81].

More recently, the RBD region of the SARS-CoV-2 has been produced in the diatom *P. tricornutum* (Table 1). The purified diatom-RBD was serologically and biologically active, was recognized by anti-RBD polyclonal antibodies, and also competitively inhibited the binding of RBD produced in mammalian cells to the human ACE2 receptor [71]. Therefore, such products derived from microalgae could provide some facilities for the construction of cheap immunoassay-based serological diagnosis kits against the SARS-CoV-2 virus [88]. For example, the RBD produced in *P. tricornutum* has been used in lateral flow devices for serological detection of IgG antibodies specific to SARS-CoV-2 in the donor sera. These diagnosis kits show the same levels of sensitivity as those previously developed with recombinant RBD produced in mammalian cells [71].

Until now, the potential of microalgae for the production of human ACE2 has not been evaluated, although some reports have identified that plants such as *N. benthamiana* could successfully produce the functional form of this glycoprotein. Indeed, ACE2 fused to the Fc domain of a human IgG1 was transiently expressed in *N. benthamiana*. Immunoblotting assays revealed that the ACE2-Fc fusion has been successfully produced and assembled in the plant cells where anti-ACE2 and anti-Fc antibodies identified a 250 kDa band in the transformants. Further data showed that the purified recombinant protein possesses the potent binding capacity to the RBD region of SARS-CoV-2, resulting in significant inhibition of the virus in vitro. Importantly, in vitro treatment of SARS-CoV-2 infected Vero cells with the plant recombinant ACE2-Fc protein significantly inhibits the virus infectivity [103]. Furthermore, a truncated version of human ACE2 was produced in a soluble form in *N. benthamiana*. The recombinant soluble ACE2 produced by the plant was able to successfully bind to the S-glycoprotein of the SARS-CoV-2 virus. In this context, microalgae have major advantages that could make them a more potent cellular biofactory for human ACE2 production. Unlike plants, some microalgae such as *P. tricornutum* contain less immunogenic epitopes on their *N*-glycans and thus it can be assumed that these microorganisms could allow the production of human glycoproteins with a lower induction of immune response in humans. The *N*-linked glycosylation pathways in *P. tricornutum* have been demonstrated to have a high similarity to those of mammalian cells [104,105,106]. Indeed, several exogenous glycosylated proteins expressed in mammalian systems are successfully produced in *P. tricornutum* with similar post-translational modifications, including *N*-glycans, confirming that the glycoproteins produced by microalgal are fully functional and not immunogenic [67,68,71,81].

## 6. Edible Microalgae as a Biosystem for Production and Delivery of S-Glycoprotein and Human ACE2

### 6.1. Edible Microalgae-Based Vaccine against SARS-CoV-2

As a novel concept, microalgae could be designated as a unique platform for the edible delivery of S-glycoprotein and human ACE2 for inhibition of the SARS-CoV-2 virus and treatment of the COVD-19 (Figure 3). It is important to note that microalgae have no common pathogens with humans and some have already been approved by the FDA as human food. Although the whole *P. tricornutum* is not approved as human food yet, its nutritional composition is highly similar to the microalga *Odontella aurita* that was recently approved by European Food Safety Authority (EFSA) [107]. These microorganisms could be grown in closed/controlled bioreactors, which precisely meet the requirements of good manufacturing practice (GMP) for the recombinant proteins produced [21]. Some microalgae (e.g., *C. reinhardtii*, *C. vulgaris*, and *P. tricornutum*) possess a rigid cell wall that provides a natural encapsulation vehicle for the delivery of recombinant biologics in oral forms (Figure 3) [21]. The cell wall of many green microalgae has a multilayered and complex structure enriched in polysaccharides, proteins, and glycoproteins. For example, the cell wall of *C. reinhardtii* has a seven-layered composition made of high molecular weight carbohydrate polymers and several hydroxyproline-rich proteins [108,109]. Similarly, *Chlorella* species have a rigid, multi-layered cell wall containing 1–2 microfibrillar and mono or trilaminar outer layers embedded in a plastic polymeric matrix [110]. Moreover, the cell wall of *P. tricornutum* has a rigid silicified structure composed of organic molecules, especially a sulfated glucuronomannan and a mannan chain decorated with a sulfate ester group [111]. Therefore, such cell wall represents the ideal formulation for the successful oral delivery of the vaccine that can overcome the harsh gastrointestinal environment. Moreover, as the microalgae cell environment has already been demonstrated to possess immunomodulatory activity, thus minimizing tolerance induction to achieve effective protection [112,113].

Edible vaccines are mucosal-targeted biologics that could simultaneously stimulate both the systematic and mucosal immune systems and activate the first line of defense of the human body through the mucosa (Figure 3) [114]. Given that many pathogens might enter the human body through the mucosal surfaces of the digestive, respiratory, or urinary reproductive systems, edible vaccine delivery could be considered as an effective approach for targeted antigens delivery [115]. It is believed that edible vaccines produced in microalgae can successfully pass the harsh condition of the stomach and release antigens in the small intestine through their cell wall digestion by acting bacterial digestive enzymes [114]. Antigen release could induce immunogenic cascades contributing to microfold cells (M cells) localized in Peyer’s patches, antigen-presenting cells (e.g., dendritic cells), helper T-cells, and B cells and subsequent production of secrete immunoglobulin G and A (IgG/IgA) [116]. Antigens taken up by M cells are transferred to the underlying immune system (i.e., Peyer’s patches) of the mucosae by transcytosis (Figure 3) [117]. Thus far, a few attempts have been made to identify the potential of *C. reinhardtii* for the production of edible vaccines against infectious diseases. The edible vaccine produced from *C. reinhardtii* can protect the antigen against *Staphylococcus aureus* when the transformed whole cells were exposed to a stomach-mimicking environment (37 °C, pH 1.7, 0.5 mg/mL pepsin). More importantly, when mice were repeatedly fed (five weeks oral vaccination) with freeze-dried microalga cells, both fecal IgA and IgG antibody titers increased against expected *S. aureus*, resulting in 80% survival of infected mice [118]. Notably, *Chlorella* species could also persist in highly acidic environments, as 6N HCl is required to hydrolyze their rigid cell wall [110]. Today, research is underway to evaluate the potential of *C. reinhardtii* for the production of edible vaccines against SARS-CoV-2 (Figure 3). An Italian research group tried to produce an edible *C. reinhardtii* expressing the RBD region of SARS-CoV-2 by adopting either nuclear, chloroplast, or both, transgenesis [119]. In addition, to the best of our knowledge, another trial is ongoing and is being carried out by the Company TransAlgae Biotech, which is currently producing an edible vaccine for the stimulation of immune responses against SARS-CoV-2. Preliminary results have shown that the addition of S-glycoprotein at appropriate concentrations does not alter the safety of the processed *C. reinhardtii* for human use [88]. Of note, lyophilized microalgae-based edible vaccines can simply be stored at room temperature for several months (up to 20 months), reducing the risk of expiration and loss of functionality of recombinant antigens or proteins [114]. Unlike plants, microalgae can produce high-quality and highly homogeneous proteins, which reduces dosing issues for edible vaccines [120].

### 6.2. Targeted Delivery of Soluble rhACE2 by Microalgae

Blocking or reducing the binding reaction of the SARS-CoV-2 S-glycoprotein to the human ACE2 has presented an alternative approach for the treatment of patients with COVID-19 [121]. Given the large size of the human ACE2 receptor (740 amino acids) and concerns about immunogenic reactions, the potential of truncated forms of soluble rhACE2 was evaluated [53]. However, truncated forms of ACE2 are unstable, have a short half-life, and degrade rapidly resulting in a diminished ability to effectively trap the virus [121]. Apart from its large size, the human ACE2 protein contains seven *N*-glycosylation sites (N53, N90, N103, N322, N432, N546, and N690) as well as four disulfide bonds in its primary structure that could directly modulate the binding reaction and affinity between ACE2 and SARS-CoV-2 S-glycoprotein [122,123]. The reduction of disulfide bonds to thiol groups significantly impairs the interaction between ACE2 and SARS-CoV-2 [124]. Due to a potent protein synthesis mechanism and a natural encapsulation possibility, microalgae could be designated as an alternative biosystem for the production and delivery of soluble rhACE2 (Figure 4). Microalgae possess the necessary enzymes and proteins to perform *N*-glycosylation and disulfide bonds formation on recombinant proteins [21,125]. The presence of a large amount of ACE2 is crucial as a co-receptor for nutrient absorption and especially for amino acid resorption [126]. Given the interface of the gastrointestinal tract with the external environment, it is hypothesized that the gut could be considered as a possible entry route for SARS-CoV-2 (Figure 4) [127]. Considering the important biological activities of ACE2 in the regulation of cardiovascular functions and innate immune responses, caution should be considered when this glycoprotein is used as a therapeutic molecule. Therefore, efforts must be made to develop a formulation capable of specifically delivering ACE2 to target tissues [121]. Microalgae can express rhACE2 and protect it from the digestive system, and ultimately delivering it to the small intestine where there is a high accumulation of SARS-CoV-2 (Figure 4). As mentioned earlier, several microalgae are considered “generally regarded as safe” (GRAS) by the FDA in order that rhACE2 could be orally delivered to target tissues without undergoing lengthy protein purification procedures (Figure 4) [21]. In this context, microalgae seem to be a perfect vehicle for the production and targeted delivery of human rhACE2 into the gastrointestinal tract upon which they could maximize the targeted therapeutic or preventive efficacy while reducing side effects. After releasing rhACE2 from microalgae cells into the small intestine, they can competitively bind to the SARS-CoV-2 virus and interfere with its entry into target cells (Figure 4). In addition, it is suggested that the use of vitamin D may partially prevent COVID-19 infection. Indeed, vitamin D appears to suppress ACE and induce ACE2 protein expression, on which it can be used to fight SARS-CoV-2 through the RAAS pathway [128]. Some microalgae, such as *C*. *reinhardtii* and *D. tertiolecta*, contain vitamin D, which might be beneficial for the treatment of COVID-19 infection [129,130]. As a result, the oral delivery of rhACE2 from transfected microalgae can simultaneously act against SARS-CoV-2 in two different ways (Figure 2 and Figure 4).

It is notable that currently, the production of proteins through the nuclear genome of microalgae presents some difficulties, such as instability and poor expression levels, which limits the production yield of therapeutic recombinant biologics [131]. Although the heterologous protein yielding is generally higher by microalgae chloroplast transformation, the absence of glycosylation machinery, forming insoluble aggregates, and intense codon bias are the main disadvantages of the chloroplastic production of complex glycoproteins [21,114]. Until now, many attempts have been made to circumvent poor gene expression levels through nuclear transformation. The use of strong native (e.g., hsp70, rbcS2, and PSAD) or synthetic (i.e., hsp70/rbcS2 and hsp70/βTUB2) promoters and native terminators (e.g., FDX1) could improve the expression efficiency of the recombinant proteins [131,132]. It is, for example, reported that the regulatory elements of the hsp70 promoter may actively counteract gene silencing during the nuclear transformation of *C. reinhardtii* [132]. Moreover, introns I, II, and III of the rbcS2 and intron I of LHCBM1 genes contained regulatory elements that helped to improve the transcription of *C. reinhardtii* nuclear transgenes [133,134]. Furthermore, 2A peptides, which originated from some viruses such as the foot and mouth disease virus, could also significantly increase transcription and translation efficiency after microalgae nuclear transformation [22]. Of note, many attempts at using synthetic biology and deep learning approaches have been performed to optimize the stable and robust production in microalgae of fully functional recombinant proteins through nuclear transformation [66,119,121].

## 7. Conclusions

Taken altogether, we believe that microalgae could be a powerful alternative biosystem for the production of S and human ACE2 glycoproteins for the treatment of patients with COVID-19 infection. In addition, these microorganisms offer a high potential for edible delivery of the produced glycoproteins, precisely into different parts of the gastrointestinal tract infected with SARS-CoV-2. This easy, fast, cheap, and simple strategy can significantly reduce the cost of COVID-19 vaccine production and ACE2 therapy per treatment course.

## Figures and Tables

**Figure 1 marinedrugs-20-00657-f001:**
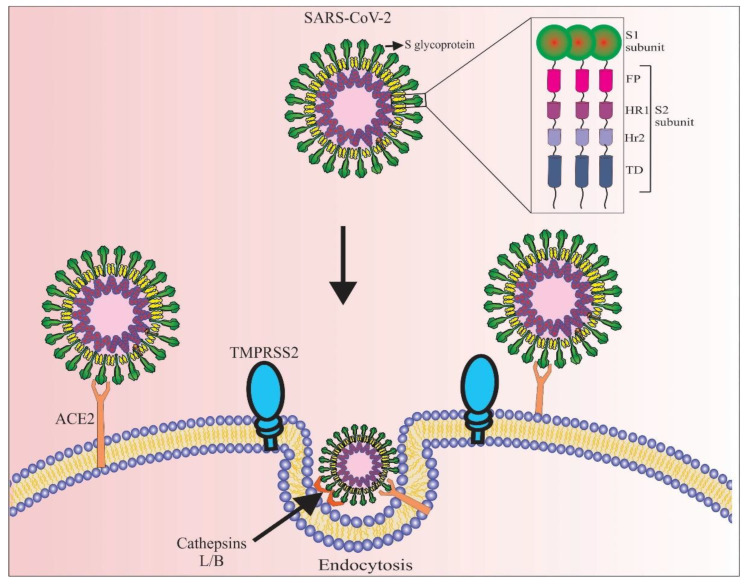
Schematic representation of the SARS-CoV-2 virus and its mechanism of entry into the human cells. The spike glycoproteins (S-glycoprotein) that cover all surfaces of SARS-CoV-2 bind to ACE2 receptors that are primarily expressed in alveolar epithelial cells in the lung and enterocytes of the small intestine. The S-glycoprotein is composed of the S1 and S2 subunits, the latter consisting of the FP, HR1, HR2, and TD domains. Upon interaction of S-glycoprotein with ACE2, TMPRSS2, and in some cases cathepsin L or B activates S-glycoprotein by proteolytic cleavage and facilitates SARS-CoV-2 virus entry by endocytosis. FP: fusion peptide domain, HR1: heptad-repeat domain I, HR2: heptad-repeat domain II, and TD: transmembrane domain.

**Figure 2 marinedrugs-20-00657-f002:**
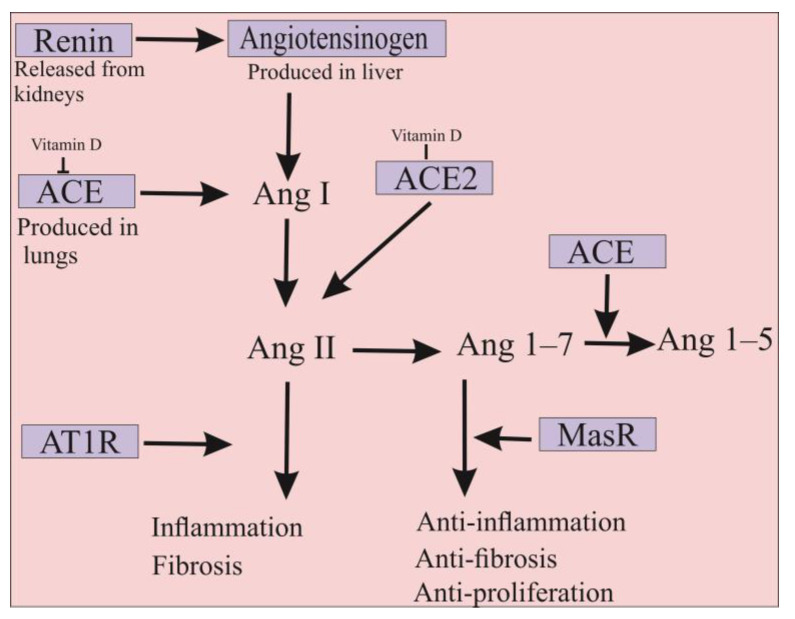
Differences between ACE2 and ACE functions. ACE is the major component of the renin-angiotensin-aldosterone system (RAAS) and is involved in the inflammation and fibrosis pathways. ACE2 is a homolog of ACE that counteracts the physiological function of ACE in order that the protein has anti-inflammation and anti-fibrosis properties. In addition, ACE2 has anti-proliferation properties that can significantly inhibit SARS-CoV-2 infection. Moreover, vitamin D is involved in the suppression of ACE and also in the increase of ACE2 expression. Therefore, the administration of this molecule could be used to prevent SARS-CoV-2 infection. Ang; angiotensin, AT1R; angiotensin II type I receptor and MasR; Mas receptor.

**Figure 3 marinedrugs-20-00657-f003:**
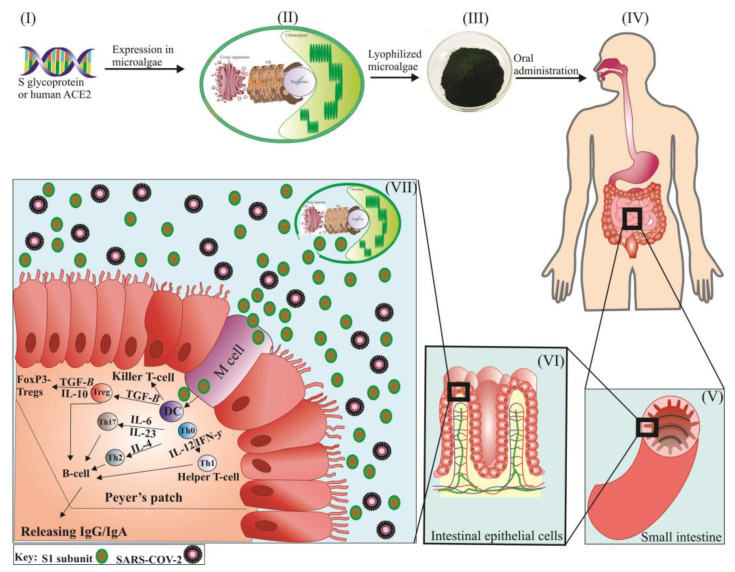
The mechanism of action of an edible microalgae vaccine against SARS-CoV-2 is composed of the S1 subunit. The S1 subunit is expressed in a soluble form by the microalgae nuclear genome (I and II). Then, the transformants are lyophilized (III) and administrated orally to patients with COVID-19 infection (IV). The transgenic microalgae can successfully pass through the gastric system (V and VI) due to their thick and rigid cell wall. Upon reaching the small intestine (one of the routes of entry for SARS-CoV-2), the cell wall of microalgae is hydrolyzed by bacterial enzymes, resulting in the release of produced recombinant antigens. The released antigens (i.e., S1 subunit) are taken up by M cells and subsequently captured by antigen-presenting cells such as DC, eliciting humoral immune responses against SARS-CoV-2 by triggering various cytokines, including Th0, Th1, Th2, Th17, IL-4, IL-6, IL-23, TFG-β, and IFN-γ. Both humoral immune responses could produce secretory IgA and IgG against the SARS-CoV-2 virus. DC: dendritic cells, Th: T helper type, IL: interleukin, TFG-β: transforming growth factor beta, IFN-γ: interferon-gamma, FoxP3: Forkhead box protein P3, Tregs: regulatory T cells.

**Figure 4 marinedrugs-20-00657-f004:**
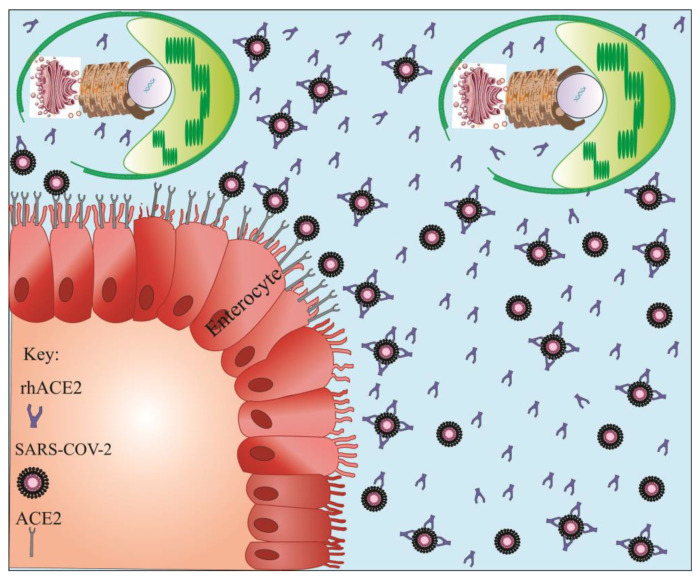
Targeted delivery of recombinant human ACE2 (rhACE2) produced by microalgae into the small intestine. Given the high expression of ACE2 in small intestine enterocytes and the presence of SARS-CoV-2 in this region, rhACE2 released from microalgal cells can competitively bind to SARS-CoV-2 and prevents virus entry into target cells. Viruses inactivated by rhACE2 can then be excreted from the human body via feces.

**Table 1 marinedrugs-20-00657-t001:** Summary of the recombinant proteins that have been produced in microalgae and that are intended to be used as biologics.

Recombinant Protein	Expression Host	Targeted Disease	Biological Activity	References
Human interleukin 2	*C. reinhardtii*, *C. vulgaris*, and *D. salina*	Cancer	Yes	[64]
Human Anti-Hepatitis B surface antigen antibody (CL4mAb)	*P. tricornutum*	Hepatitis B	Yes	[65,66]
Hepatitis B surface antigen	*P. tricornutum*	Hepatitis B	Yes	[65]
RBD of SARS-CoV-2	*P. tricornutum*	COVID-19	Yes	[71]
Human growth hormone	*C. reinhardtii*	Turner syndrome	Yes	[77]
Hepatitis B surface antigen	*D. salina*	Hepatitis B	Yes	[78]
Human interferon alpha 2a	*C. reinhardtii*	Cancer	Yes	[79]
Spike glycoprotein of SARS-CoV-2	*C. reinhardtii*	COVID-19	Yes	[80]
RBD of SARS-CoV-2	*C. reinhardtii*	COVID-19	Yes	[81]
BCB; a multi-epitope protein	*Schizochytrium* sp.	Breast cancer	Yes	[82]
Human VEGF-165	*C. reinhardtii*	Wound healing	Yes	[83]

RBD: receptor binding domain; Human VEGF-165: human vascular endothelial growth factor A.

## Data Availability

Not applicable.

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
