# Peer review of "Microalgae as an Efficient Vehicle for the Production and Targeted Delivery of Therapeutic Glycoproteins against SARS-CoV-2 Variants"

_marinedrugs, 2022, doi:10.3390/md20110657_

Round 1

Reviewer 1 Report

Reviewer #1: 

Coronavirus disease 2019 (COVID-19) has emerged as a new world pandemic, infecting millions of people with a substantial mortality. Therapeutic alternatives are crucial to control pandemic situation. Therefore, there is significant interest in understanding new strategies against SARS-CoV-2 as are Microalgae.

Recently publications show several alternative therapeutic and vaccine candidates in Clinical trials and this review could be help in understand this pandemic.  

In this manuscript, by Jaber Dehghani et al titled “Microalgae as an efficient vehicle for the production and targeted delivery of therapeutic glycoproteins against SARS-CoV-2 variants”.

The authors performed a description of as an efficient protein synthesis machinery and the ability to properly impose post-translational modifications make microalgae an eco-friendly platform for the production of pharmaceutical glycoproteins.

And mentioned as several microalgae (e.g., Chlamydomonas reinhardtii, Dunaliella bardawil, and Chlorella species) are already approved by U.S. FDA as safe human food. 

There are several concerns that to be addressed.

This manuscript is well written and sites key findings in the field, therefore it will be helpful for investigators entering into coronavirus therapeutic-vaccine/COVID-19 research. The study would benefit the section on general aspects concern to COVID-19 disease. Comments to improve the clarity of the manuscript are provided below.

Comments for the authors' consideration:

1.     Please add a section explaining how the production of recombinant S-glycoprotein could be moderate the immune response and if the authors know main epitopes relevant with your strategies. 

2.     To elaborate a table showed treatment of patients with other infections or pathologies 

used microalgae.

3.     Add a paragraph mentioned the limitations of this methodology. 

Author Response

Dear Editor,

Please find enclosed the revised version of our manuscript as well as the detailed answers to the reviewer comments (point to point response to the reviewer’s comments below). We hope to have addressed all of them and clarify the reviewer’s concerns.

We are looking forward to hear for your final decision,

Best regards

Pr Muriel Bardor

University of Rouen Normandy, GlycoMEV lab

Point to point response to the reviewer’s comments.

Dear reviewer #1

Thanks for reviewing our manuscript and your valuable comments to improve the quality of our work. We acknowledge the issues and suggestions given and have amended appropriate changes on the manuscript. Followings are the details of our correctness based on your comments. We hope that our corrections and explanations satisfy your comments.

Yours sincerely,

Authors

Comment:

  1. Please add a section explaining how the production of recombinant S-glycoprotein could be moderate the immune response and if the authors know main epitopes relevant with your strategies.

Author’s response:

-Thanks for your valuable suggestion. Explanations regarding the immune response were added in the revised manuscript lines 163-173. Moreover, we also modified the manuscript lines 481-485 towards the direction given by the comment.

  1. To elaborate a table showed treatment of patients with other infections or pathologies

used microalgae.

Author’s response:

-Based on this comment and in order to complete the manuscript, we introduced in its revised version the Table 1 summarizing all the recombinant proteins that have been produced in microalgae and are intended to be used for therapeutic purpose.

  1. Add a paragraph mentioned the limitations of this methodology.

Author’s response:

- A paragraph mentioning the actual limitations of using microalgae cells as an expression system has been included in the revised version of the manuscript lines 583-602.

Reviewer 2 Report

- Please add a sentence in the abstract stating the main goal/focus of this revision. 

- Throughout the manuscript: Italicize “in vivo” and “in vitro

- line 129: please refer Figure 1 after “….1273 residues)”.

- lines 150-152: This sentence may lead to misinterpretations. It is not the S-glycoprotein and ACE2 that may be effective in the inhibition of COVID-19 infection, but rather the targeting of these proteins for therapeutical approaches, including their recombinant production and delivery in vaccines. Please rephrase.

- Fig. 1 legend: Please replace “structure of SARS-CoV-2 virus” by “schematic representation of the SARS-CoV-2 virus”.

- line 203: The major component of the renin-angiotensine-aldosterone system is ACE and not ACE2. Please correct.

- lines 237-239: Please replace “deduced” by hypothesized.

- lines 264-265: Please replace “identified” by produced.

- line 267: Please describe/contextualize what is “Ang 1-5” or add it in Figure 2.

- line 295 – This entire section is focused on ACE2. Therefore, the It does not make much sense to mention the s-glycoprotein in this line.

- lines 323, 340-345: Although commonly known as microalgae, Spirulina platensis is a cyanobacteria and, thus, a prokaryote. Therefore, it should not be mentioned here.

- lines 357: 359: Already stated in lines 78-80.

- line 381: Replace “further data identified ..” by “further data showed”.

- line 383:The study mentioned showed that S-glycoprotein produced by microalgae could efficiently bind to ACE2 using recombinant ACE2 or cells expressing ACE2. In both cases, the assays are in vitro. Please rephrase.

- line 387: Italicize “Agrabacterium“.

- lines 419-429: This section describes the advantages of using microalgae for ACE2 production. This should be integrated into a previous paragraph (lines 368-378) that already discusses this idea.

- line 430: Replace “S and human ACE2 glycoproteins” by S-glycoprotein and human ACE2

- line 435: Replace “..and have already been approved” by “..and some have already been approved”

- line 448: Italicize Chlorella

- lines 458 – 460: The sentence is a bit confusing. Do you mean that the vaccines produced by microalgae can pass the harsh conditions of the stomach and release the antigens in the small antigen upon the activity of the bacterial digestive enzymes? Please rephrase.

- lines 517-519: Already stated in lines 222-223.

Author Response

Dear reviewer #2

Thanks for reviewing our manuscript and your valuable comments to improve the quality of our work. We acknowledge the issues and suggestions given and have amended appropriate changes on the manuscript. Followings are the details of our correctness based on your comments. We hope that our corrections and explanations satisfy your comments.

Yours sincerely,

Authors

Comment:

- Please add a sentence in the abstract stating the main goal/focus of this revision. 

Author’s response:

- Thanks for your valuable suggestion. Your comment has been considered in the revised manuscript and a new sentence has been added to lines 36-38.

Comments:

- Throughout the manuscript: Italicize “in vivo” and “in vitro

- line 129: please refer Figure 1 after “….1273 residues)”.

Author’s response:

- We considered these comments in the revised manuscript and made the necessary arrangements.

Comment:

- lines 150-152: This sentence may lead to misinterpretations. It is not the S-glycoprotein and ACE2 that may be effective in the inhibition of COVID-19 infection, but rather the targeting of these proteins for therapeutical approaches, including their recombinant production and delivery in vaccines. Please rephrase.

Author’s response:

-Thanks for your suggestions. We rephrased the sentence in the revised manuscript lines 157-159. We hope that it read better now.

Comment:

- Fig. 1 legend: Please replace “structure of SARS-CoV-2 virus” by “schematic representation of the SARS-CoV-2 virus”.

- line 203: The major component of the renin-angiotensine-aldosterone system is ACE and not ACE2. Please correct.

- lines 237-239: Please replace “deduced” by hypothesized.

- lines 264-265: Please replace “identified” by produced.

- line 267: Please describe/contextualize what is “Ang 1-5” or add it in Figure 2.

- line 295 – This entire section is focused on ACE2. Therefore, the It does not make much sense to mention the s-glycoprotein in this line.

Author’s response:

-All these comments have been taken into consideration and have been included in the revised manuscript.

Comment:

- lines 323, 340-345: Although commonly known as microalgae, Spirulina platensis is a cyanobacteria and, thus, a prokaryote. Therefore, it should not be mentioned here.

Author’s response:

-Thanks for your valuable suggestion. Many published articles presented Spirulina platensis as a microalga. To the best of our knowledge, plant biologists have utilized blue microalga, and microbiologists utilized cyanobacterium for introducing this organism. However, based on your comment, the sentence referring to Spirulina platensis has been removed in the revised version of the manuscript.

Comments:

- lines 357: 359: Already stated in lines 78-80.

- line 381: Replace “further data identified ..” by “further data showed”.

- line 383:The study mentioned showed that S-glycoprotein produced by microalgae could efficiently bind to ACE2 using recombinant ACE2 or cells expressing ACE2. In both cases, the assays are in vitro. Please rephrase.

- line 387: Italicize “Agrabacterium“.

- lines 419-429: This section describes the advantages of using microalgae for ACE2 production. This should be integrated into a previous paragraph (lines 368-378) that already discusses this idea.

- line 430: Replace “S and human ACE2 glycoproteins” by S-glycoprotein and human ACE2

- line 435: Replace “..and have already been approved” by “..and some have already been approved”

- line 448: Italicize Chlorella

- lines 458 – 460: The sentence is a bit confusing. Do you mean that the vaccines produced by microalgae can pass the harsh conditions of the stomach and release the antigens in the small antigen upon the activity of the bacterial digestive enzymes? Please rephrase.

- lines 517-519: Already stated in lines 222-223.

Author’s response:

-Thanks for your valuable suggestions. We considered all the comments in the revised manuscript.